# Species Survey of Leaf Hyponasty Responses to Warming Plus Elevated CO_2_

**DOI:** 10.3390/plants13020204

**Published:** 2024-01-11

**Authors:** Michael D. Thomas, Reagan Roberts, Scott A. Heckathorn, Jennifer K. Boldt

**Affiliations:** 1Department of Environmental Sciences, University of Toledo, Toledo, OH 43606, USA; reagan.roberts@rockets.utoledo.edu; 2Agricultural Research Service, United States Department of Agriculture (USDA), Toledo, OH 43606, USA; jennifer.boldt@usda.gov

**Keywords:** climate change, hyponasty, leaf angle, morphology

## Abstract

Atmospheric carbon dioxide (CO_2_) concentrations are increasing and may exceed 800 ppm by 2100. This is increasing global mean temperatures and the frequency and severity of heatwaves. Recently, we showed for the first time that the combination of short-term warming and elevated carbon dioxide (eCO_2_) caused extreme upward bending (i.e., hyponasty) of leaflets and leaf stems (petioles) in tomato (*Solanum lycopersicum*), which reduced growth. Here, we examined additional species to test the hypotheses that warming + eCO_2_-induced hyponasty is restricted to compound-leaved species, and/or limited to the Solanaceae. A 2 × 2 factorial experiment with two temperatures, near-optimal and supra-optimal, and two CO_2_ concentrations, ambient and elevated (400, 800 ppm), was imposed on similarly aged plants for 7–10 days, after which final petiole angles were measured. Within Solanaceae, compound-leaf, but not simple-leaf, species displayed increased hyponasty with the combination of warming + eCO_2_ relative to warming or eCO_2_ alone. In non-solanaceous species, hyponasty, leaf-cupping, and changes in leaf pigmentation as a result of warming + eCO_2_ were variable across species.

## 1. Introduction

Rising atmospheric carbon dioxide (CO_2_) concentrations are causing global mean temperatures to increase, leading to more frequent and severe heatwaves [1]. Plants must adapt to climate change through chemical or structural changes to maintain productivity [2,3]. Benefits of elevated CO_2_ include increased net photosynthesis through the reduction in photorespiration and increased water-use efficiency from reduced stomatal conductance [4,5]. Elevated CO_2_ (eCO_2_) concentrations may affect the ability of plants to cope with heat stress via changes in sensitivity to, or concentrations of, plant hormones [6,7,8,9]. Adaptation to multiple concurrent climate change factors like warming + eCO_2_ will likely have unpredictable, and possibly severe, consequences for plant growth [10,11,12,13].

In a previous study, the combination of short-term warming + eCO_2_ (37 °C and 700 ppm) increased upward petiole angle (hyponasty) more than either factor alone in tomato (*Solanum lycopersicum*; ‘Early Girl’, ‘Brandywine’, and H3406) [14]. Further evaluation of the effects of warming + eCO_2_-induced hyponasty demonstrated that the severity of petiole hyponasty increases with temperature at eCO_2_. CO_2_ concentrations above ambient (600, 800, 1000 ppm) resulted in similarly increased leaf angles at increased temperature, suggesting that eCO_2_ has an additive effect on the previously described thermal hyponasty in tomato plants [15]. Thermal hyponasty can be grouped into a suite of plant responses to high temperatures called thermomorphogenesis. Thermomorphogenesis also includes other morphological changes in response to increased temperatures, such as elongation of the hypocotyl, petiole, and/or root; leaf or petiole hyponasty; reduced leaf thickness or stomatal density; changes in leaf area; and earlier or later flowering [16,17,18]. These morphological responses are often the result of thermal deactivation of the light-sensing compound phytochrome B, resulting in increased activity of PHYTOCHROME INTERACTING FACTOR 4 (PIF) and the subsequent up-regulation of genes associated with auxin and brassinosteroid production, which lead to changes in plant architecture [16,19,20,21].

As atmospheric CO_2_ and temperatures increase, plants will experience warming + elevated CO_2_ concurrently. As a result, growth responses like warming + eCO_2_-induced hyponasty in tomatoes may occur among other species of plants and affect global plant productivity. Jayawardena [14] evaluated only a few species for their response to the additive effects of warming + eCO_2_. Their work suggested that simple-leaved plants may not exhibit dramatic hyponasty when exposed to the combination of warming + eCO_2_. Thus, a further survey of plant responses to the combined effects of warming + eCO_2_ is warranted. Our objective was to survey several additional species for leaf morphological responses (e.g., hyponasty, epinasty (downward bending), or leaf curling) to the combined influence of warming + eCO_2_ to determine the prevalence of warming + eCO_2_-induced leaf morphology change. As part of this survey, we examined both simple- and compound-leaf species, especially within Solanaceae, to test if simple-leaf plants exhibit reduced hyponasty with warming + eCO_2_ compared to compound-leaf plants.

## 2. Results

Among the solanaceous plants surveyed, leaf angle increased the most with warming + eCO_2_ in compound-leaf species, but not in simple-leaf species (Figure 1). Compared to the control, tomato (*Solanum lycopersicum* ‘Celebrity’) and potato (*Solanum tuberosum* ‘Red Pontiac’) displayed petiole hyponasty with warming (38 °C vs. 30 °C tomato, 32 °C vs. 24 °C in potato), and even greater petiole hyponasty with warming + eCO_2_ (800 ppm). Eggplant (*Solanum melongena*) displayed equally increased leaf angle with warming (38 °C) and the combination of warming + eCO_2_. Individually, warming (38 °C) and eCO_2_ increased petiole angle in pepper (*Capsicum annuum* ‘Procraft F1’) compared to both the control (30 °C) and warming + eCO_2_. Husk cherry (*Physalis pruinosa* ‘Goldie’) only displayed increased petiole angle in response to eCO_2_.

Non-solanaceous plants exhibited variable responses to eCO_2,_ warming, and warming + eCO_2_ (Figure 2). Hibiscus (*Hibiscus rosa-sinensis*) displayed some increase in petiole hyponasty in response to warming (37 °C), eCO_2_, and the combination of warming + eCO_2_, though not significantly (*p* > 0.05) when compared to the control (30 °C). Warming + eCO_2_ increased leaf angle (Figure 2) and caused leaf cupping (Figure 3) in bush bean (*Phaseolus vulgaris* ‘Affirmed’). Nasturtium (*Tropaeolum majus* ‘Empress of India’) displayed dramatic changes in leaf angle with the combination of warming (28 °C vs. 22 °C) + eCO_2_. Spearmint (*Mentha* × *piperita*) leaves displayed a similar epinastic (downward curling of leaves) response to control and warming + eCO_2_ but had reduced epinasty with warming alone (Figure 3). Rose (*Rosa multiflora* ‘Oso Easy Double Red’) had highly curled leaves in response to warming + eCO_2_ compared to all other treatments and exhibited a downward bending of leaves under warming + eCO_2_ (visual observation).

Leaf color visually changed in response to treatments in pepper, hibiscus, mint, and nasturtium (Figure 4). The leaves of pepper and hibiscus plants grown at eCO_2_ were lighter green, regardless of the temperature treatment. Mint leaves turned light green with purple-brown patches in eCO_2_ conditions, regardless of the temperature treatment. Exposure to warming + eCO_2_ in nasturtium resulted in a darker leaf and petiole color compared to all other treatments.

All plants accumulated less biomass (ground cherry not collected) under warming treatment compared to the control, with variable responses seen at eCO_2_ or warming + eCO_2_ (Figure 5).

## 3. Discussion

To our knowledge, fourteen species have been surveyed to date for warming + eCO_2_-induced hyponasty: six compound-leaf species and eight simple-leaf species ([14] and this study). All compound-leaf plants surveyed displayed hyponasty and/or leaf cupping in response to warming + eCO_2_. In contrast, most simple-leaf plants displayed no hyponastic response to warming + eCO_2_. Five of the eight simple-leaf plants surveyed in this study had no noticeable change in leaf angle with warming and/or eCO_2_; of the three species that did display changes in leaf angle, nasturtium was the only one with significantly increased leaf angle under warming + eCO_2_. This could be due to nasturtium’s leaf structure, as it was the only peltate-leaved plant to be surveyed. Interestingly, hibiscus, one of two simple-leaf malvids examined, displayed a visible hyponastic response, though the change in leaf angle was not significant, likely due to the low number of replicates (n = 3). Eggplant, a simple-leaf solanaceous crop, displayed similar increases in leaf angle under warming and warming + eCO_2_ treatments.

Phenotypic responses in these two studies seem to be mostly determined by leaf architecture, though some trends may be present within plant families. Both compound-leaf legumes surveyed (bush bean and soybean) displayed small increases in both petiole angle and leaf cupping with warming + eCO_2_. Compound-leaf species surveyed in Solanaceae (potato and tomato) and Asteraceae (marigold) displayed hyponasty in response to warming + eCO_2_, while simple-leaf species (pepper, ground cherry, and sunflower) did not. Eggplant displayed petiole hyponasty under warming conditions regardless of CO_2_ concentration. Hibiscus and okra, both simple-leaf members of Malvaceae, displayed different responses to warming and/or eCO_2_, although this may be the result of differences in leaf morphology, as hibiscus has entire leaves and okra has palmately lobed leaves.

As a consequence of leaf diversity, plants may vary greatly in warming + eCO_2_-induced phenotypic changes, making it difficult to predict the effects of climate change on plant productivity. Though not explored here, wild plants may exhibit more diverse phenotypic responses than cultivated varieties, as populations of wild plants may have larger genetic diversity, which could increase phenotypic plasticity [22]. Regardless of domestication status or growth form, plants often suffer declines in nitrogen uptake or concentration as a result of warming + eCO_2_ [12]. Additionally, leaf carotenoid content is reduced with eCO_2_, though this could be offset by other stressors like warming or drought, which stimulate carotenoid synthesis [23,24]. These changes in nitrogen and carotenoids may visibly affect leaf coloration, and thus may explain the changes in leaf color seen in pepper, hibiscus, mint, and nasturtium [25,26].

This survey is far from exhaustive and has not included crassulacean acid metabolism (CAM) or C4 plants, gymnosperms, other lower-plant lineages, non-domesticated plants, or much of the diversity of leaf structures found in plants. As such, future studies should explore the effects of concurrent stressors on other plant lineages, including non-domesticated plants, to increase our understanding of climate-change-induced phenotypic changes such as the ones described here. The consequences of such phenotypic changes may be small to dramatic, as in bush bean or tomato, respectively, and could affect crop productivity in cultivated plants. Concurrent stressors are likely to become more common and severe due to climate change and, as a result, maintaining or improving crop productivity as the world population grows will be difficult. The development of multi-stressor-tolerant cultivars will be imperative for maintaining future crop productivity, and this may include consideration of the effects of climate change on leaf morphology.

## 4. Conclusions

The combination of global warming and increasing atmospheric CO_2_ (eCO_2_) may have unexpected consequences on plant growth and productivity. For example, in this study, we report increases in leaf hyponasty (upward bending of leaf blades or petioles) in some species, or changes in other aspects of leaf growth, with warming + eCO_2_ compared to either factor alone. Increases in leaf hyponasty with climate change could negatively affect the productivity of wild or agricultural plants. Further surveys of plant responses to warming + eCO_2_ stress will be beneficial for understanding plant morphological responses to climate change.

## 5. Materials and Methods

### 5.1. Growing Conditions

Pepper (*Capsicum annuum* ‘Procraft F1’), bush bean (*Phaseolus vulgaris* ‘Affirmed’), ground cherry (*Physalis pruinose* ‘Goldie’), tomato (*Solanum lycopersicum* ‘Celebrity’), eggplant (*Solanum melongena* ‘Nadia F1’), and potato (*Solanum tuberosum* ‘Red Pontiac’) were germinated in trays containing a calcite clay–soil mixture (2:1, *v*:*v*) in a greenhouse (ca. 28 °C day/24 °C night air temperatures with a 14 h photoperiod) and watered daily with tap water. After emergence, seedlings were provided with half-strength nutrient solution weekly until transplant. Seedlings were transplanted at the “first adult leaf” stage into pots filled with the calcite clay–soil mixture. Hibiscus (*Hibiscus rosa-sinensis*), mint (*Mentha × piperita*), rose (*Rosa multiflora* ‘Oso Easy Double Red’), and nasturtium (*Tropaeolum majus* ‘Empress of India’) were purchased from local greenhouses and transplanted as needed. All plants were fertilized once to three times weekly, based on size, with full-strength nutrient solution until ready for treatment. Nutrient concentrations of the full-strength solution were 4.5 mM KNO_3_, 0.5 mM NH_4_NO_3_, 2 mM KH_2_PO_4_, 2 mM CaCl_2_, 1 mM MgSO_4_, 50 μM Fe-EDTA (ethylene-diamine-tetra-acetic acid), 50 μM H_3_BO_3_, 10 μM MnCl_2_, 5 μM ZnSO_4_, and 0.1 μM NaMoO_4_; pH = 6.0. Sets of plants (n ≥ 3) of similar size and/or growth stage (e.g., number of leaves, initiation of flowering) were then selected for treatments and transferred to growth chambers (model E36HO, Percival Scientific Inc., Perry, IA, USA).

Plants were acclimated to growth chamber environments over a 24 h period under 600 µmol m^−2^ s^−1^ photosynthetically active radiation (PAR) at the top of the plant canopy, with a near-optimal growth temperature (ca. 22 °C day/17 °C night for cool-season and 30 °C day/25 °C night for warm-season plants), a 14 h photoperiod, and ambient CO_2_ (400 ppm). Elevated-temperature (6–8 °C day/night above optimal) treatments were gradually imposed on plants by incrementally increasing day/night temperatures over 4 days to avoid heat shock. After reaching treatment temperatures, eCO_2_ treatments (800 ppm, clean-grade CO_2_ (supplied by Airgas, Toledo, OH, USA)) were started, and plants were grown for 7–10 days under treatment conditions.

### 5.2. Treatments

A 2 × 2 factorial design was used, consisting of two levels of CO_2_ (400 and 800 ppm) and two temperature settings (near-optimal and heat stress; see Table 1). The upper limit of the recommended temperature range for each species, as suggested on the packaging or the distributor’s website, was used as the near-optimal temperature and heat stress was set at 6–8 °C above near optimum to simulate a natural short-term warming event, as in [14,15].

### 5.3. Measurements

The petiole angle of young but fully matured leaves (3rd–5th most recently developed) was quantified at midday using a protractor base aligned with the stem of the plant. The resulting measure was subtracted from 90° to determine the angle of the leaf. The peltate leaves of nasturtium were measured relative to horizontal. Leaf curl index (LCI) was recorded as the measurement of the flattened leaf (L_f_) divided by the measurement of the curled leaf (L_c_), then minus 1 (LCI = L_f_/L_c_ − 1). This resulted in a value of 0 for completely flat leaves and higher values for curled leaves. Measurements from three leaves were averaged to obtain a value for each plant at the halfway point and at the end of treatment. Plants were photographed at the end to document treatment effects. Unless otherwise noted, plants were harvested upon completion of the treatments to determine shoot dry mass. Entire above-ground mass was harvested and dried for 3 days at 75 °C in plant-drying oven then weighed.

### 5.4. Data Analysis

Data were analyzed and tested for normality, equal variance, and independence with S-L, normal Q-Q, and residual vs. fitted plots, respectively, using R version 4.1.2 (R Core Team (2021), Vienna, Austria). Each species was analyzed separately. If assumptions of normality, equal variance, and independence were met, two-way analysis of variance (ANOVA) was conducted, followed by Tukey’s post hoc test if two-way ANOVA results were significant (*p* < 0.05). Leaf curl data had unequal variance, so the Kruskal–Wallace test and Dunn’s post hoc comparison were used for analysis instead of two-way ANOVA. Data were graphed using Sigma Plot version 14.5 (Systat Software, San Jose, CA, USA).

## Figures and Tables

**Figure 1 plants-13-00204-f001:**
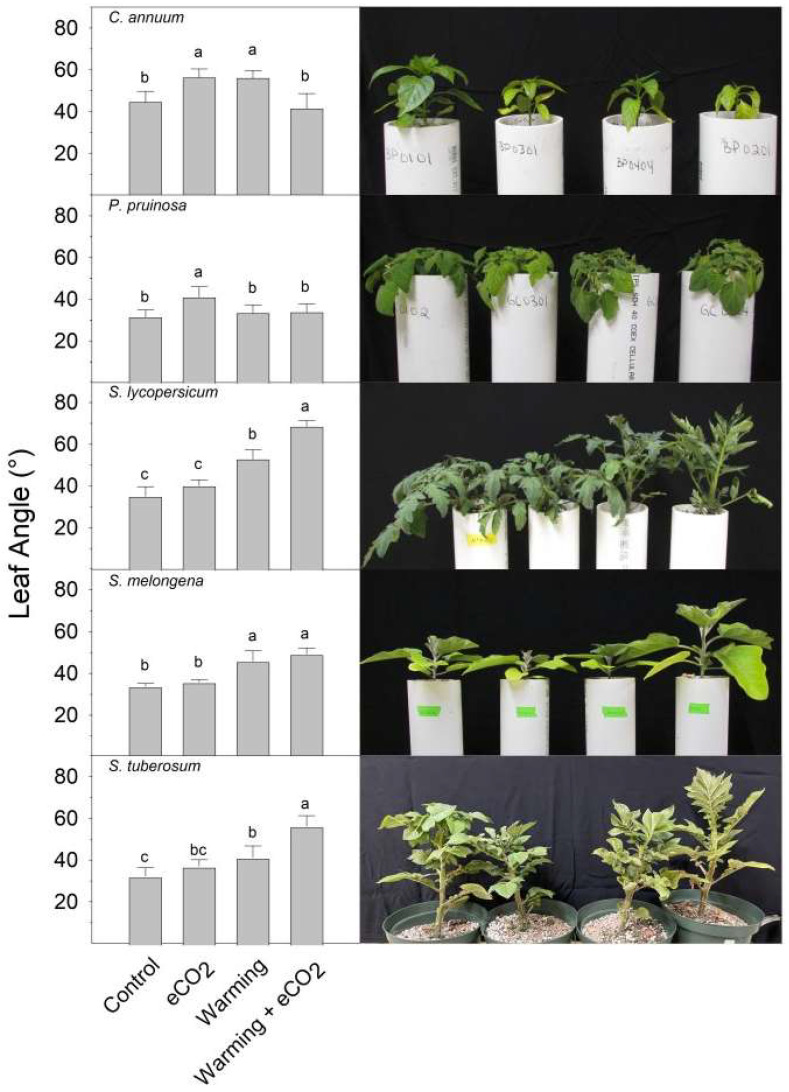
Average petiole angle in solanaceous crops, by species and treatment. From top to bottom: pepper (*C*. *annuum*), ground cherry (*P. pruinosa*), tomato (*S. lycopersicum*), eggplant (*S. melongena*), and potato (*S. tuberosum*). Plants were grown for 7 days under control (optimal temperature and 400 ppm CO_2_), warming (8 °C over optimal temperature and 400 ppm CO_2_), eCO_2_ (optimal temperature and 800 ppm CO_2_), or warming + eCO_2_ conditions (8 °C over optimal temperature and 800 ppm CO_2_). Results are means + 1 SD, with significant differences among treatments indicated by different letters above bars within each graph.

**Figure 2 plants-13-00204-f002:**
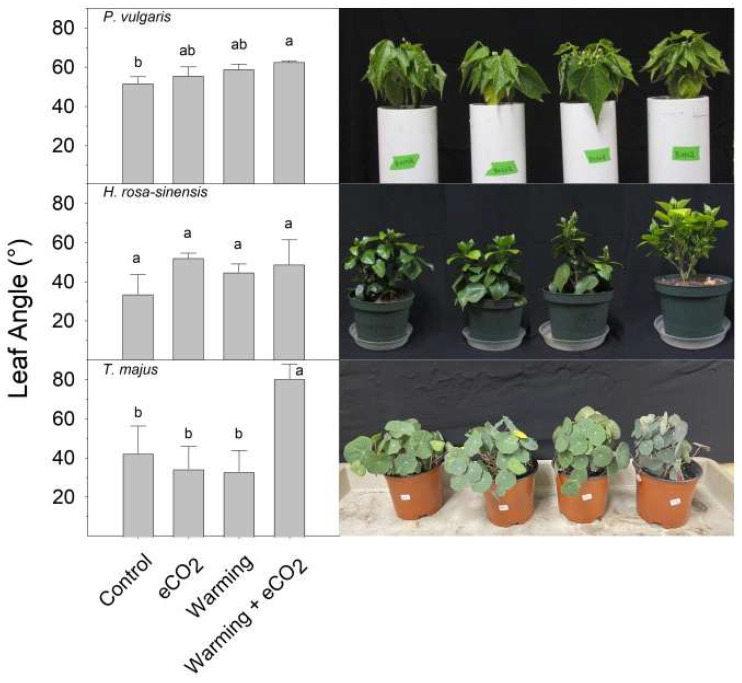
Average petiole angle in non-solanaceous plants, by species and treatment. From top to bottom: bush bean (*P. vulgaris*), hibiscus (*H. rosa-sinensis*), and nasturtium (*T. majus*). Plants were grown for 7 days under control (optimal temperature and 400 ppm CO_2_), warming (8 °C over optimal temperature and 400 ppm CO_2_), eCO_2_ (optimal temperature and 800 ppm CO_2_), or warming + eCO_2_ conditions (8 °C over optimal temperature and 800 ppm CO_2_). Results are means + 1 SD, with significant differences among treatments indicated by different letters above bars within each graph.

**Figure 3 plants-13-00204-f003:**
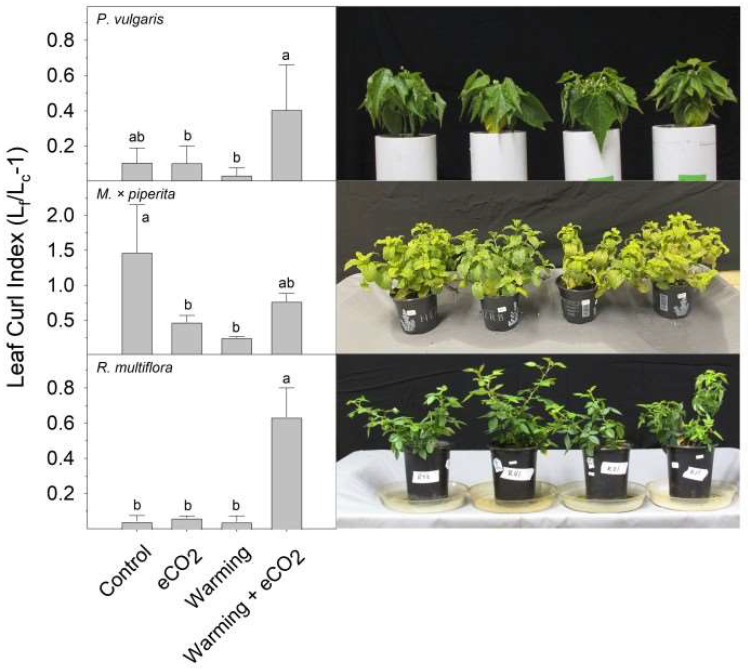
Leaf curving or leaf cupping index of species (the ratio of flattened leaf length to curled leaf length, minus 1). From top to bottom: bush bean (*P. vulgaris*), mint (*M*. *× piperita*), and rose (*R. multiflora*). Plants were grown for 7 days under control (optimal temperature and 400 ppm CO_2_), warming (8 °C over optimal temperature and 400 ppm CO_2_), eCO_2_ (optimal temperature and 800 ppm CO_2_), or warming + eCO_2_ conditions (8 °C over optimal temperature and 800 ppm CO_2_). Results are means + 1 SD, with significant differences among treatments indicated by different letters above bars within each graph.

**Figure 4 plants-13-00204-f004:**
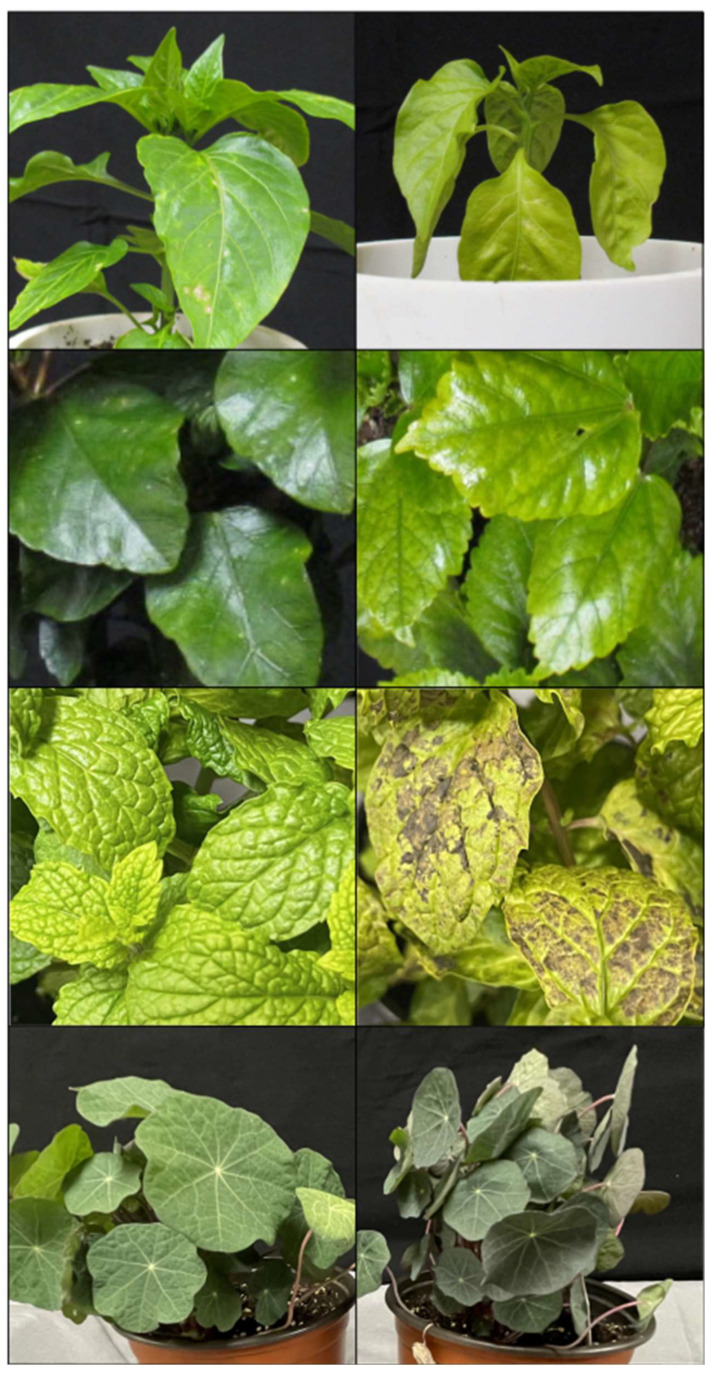
Visual comparison of leaf pigmentation under control (optimal temperature and 400 ppm CO_2_; (**left**)) and warming + eCO_2_ (8 °C over optimal temperature and 800 ppm CO_2_; (**right**)) treatments. From top to bottom: pepper (*C*. *annuum*), hibiscus (*H. rosa-sinensis*), mint (*M*. *× piperita*), and nasturtium (*T. majus*).

**Figure 5 plants-13-00204-f005:**
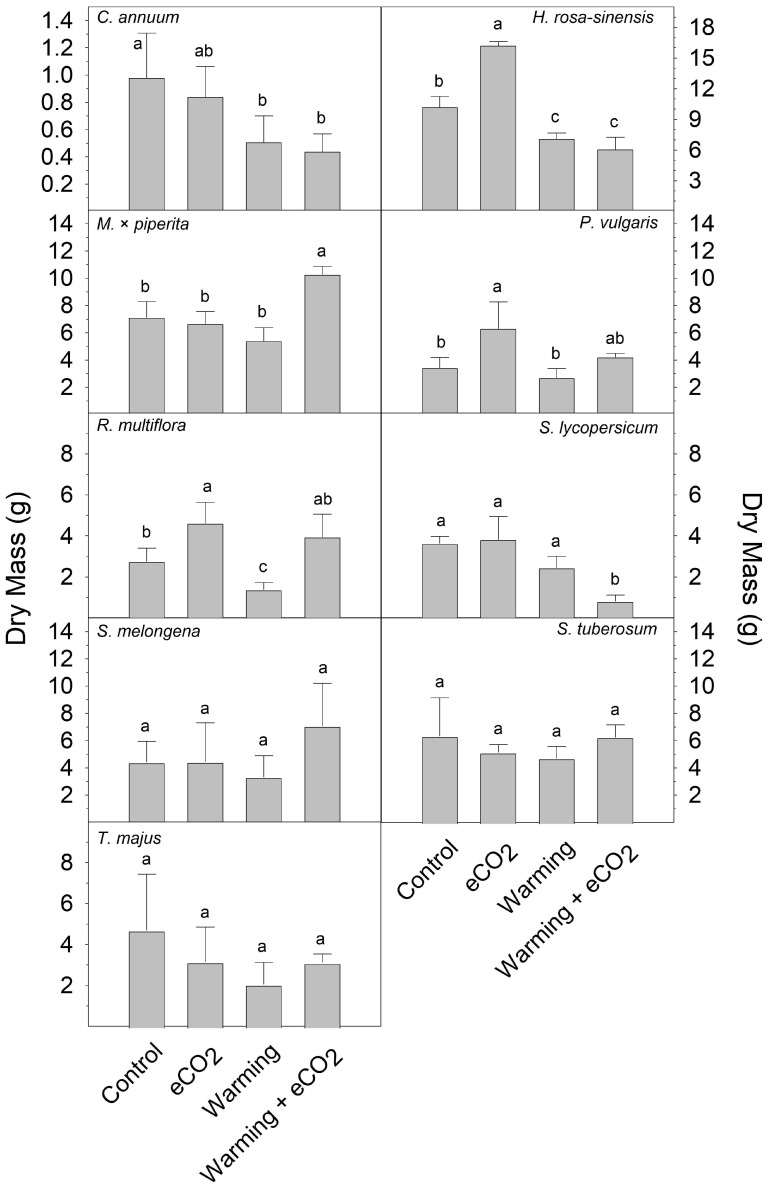
Shoot biomass of plants grown for 7 days under control (optimal temperature and 400 ppm CO_2_), warming (8 °C over optimal temperature and 400 ppm CO_2_), eCO_2_ (optimal temperature and 800 ppm CO_2_), or warming + eCO_2_ conditions (8 °C over optimal temperature and 800 ppm CO_2_): pepper (*C. annuum*), hibiscus (*H. rosa-sinensis*), mint (*M. × piperita),* bush bean (*P. vulgaris*), rose (*R. multiflora*), tomato (*S. lycopersicum*), eggplant (*S. melongena*), potato (*S. tuberosum*), and nasturtium (*T. majus*). Ground cherry (*P. pruinosa*) biomass not collected. Results are means + 1 SD, with significant differences among treatments indicated by different letters above bars within each graph.

**Table 1 plants-13-00204-t001:** Species surveyed for leaf morphology changes, including species name, family, leaf type, near optimal growth temperature treatment (day/night), and supraoptimal temperature treatment (day/night).

Species	Family	Leaf Type	Optimal Temperature (Day/Night)	Supraoptimal Temperature (Day/Night)
Pepper (*C. annuum*)	Solanaceae	Simple	28 °C/23 °C	36 °C/31 °C
Hibiscus (*H. rosa-sinesnis)*	Malvaceae	Compound	30 °C/25 °C	37 °C/32 °C
Mint (*M. × piperita*)	Lamiaceae	Simple	22 °C/17 °C	28 °C/23 °C
Bush bean (*P. vulgaris*)	Fabaceae	Compound	29 °C/24 °C	37 °C/32 °C
Ground cherry (*P. pruinosa*)	Solanaceae	Simple	30 °C/25 °C	38 °C/33 °C
Rose (*R. multiflora*)	Rosaceae	Compound	30 °C/25 °C	38 °C/33 °C
Tomato (*S. lycopersicum*)	Solanaceae	Compound	30 °C/25 °C	38 °C/33 °C
Eggplant (*S. melongena*)	Solanaceae	Simple	30 °C/25 °C	38 °C/33 °C
Potato (*S. tuberosum*)	Solanaceae	Compound	24 °C/19 °C	32 °C/29 °C
Nasturtium (*T. majus*)	Tropaeolacaeae	Peltate	22 °C/17 °C	28 °C/23 °C

## Data Availability

Data available in a publicly accessible repository. The data presented in this study are openly available in [Github] at [https://doi.org/10.5281/zenodo.10480815].

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
