# Peer review of "Species Survey of Leaf Hyponasty Responses to Warming Plus Elevated CO_2"

_plants, 2024, doi:10.3390/plants13020204_

Round 1
Reviewer 1 Report
Comments and Suggestions for Authors
The MS investigated responses of leaf hyponasty to warming and elevated CO2 and reported an interesting result.
1. The introduction should be improved as it does not state clearly why to conduct this experiment.
2. Line 180: the temperature was increased by 6-8 ℃. What's the reason for this design?
3. A 2×2 factorial design was used in this experiment, so two-way ANOVA should be used to analyze the data.
Author Response
Thank you for your review. We have made changes to our manuscript which we feel should sufficiently address your concerns.

Reviewer 2 Report
Comments and Suggestions for Authors
General comment for authors
For more than a century, it has been known that elevated atmospheric [CO2] in the air leads to enhanced plant growth because CO2 is the substrate of photosynthesis. Shortly after this awareness, ‘CO2 fumigation’ has been occasionally used to increase crop production in greenhouses. In consequence, the number of publications on this has grown exponentially since then, motivated further by deep concern about potential impacts on physiological processes in plants. In principle, also higher temperatures can stimulate growth of plants up to an optimum of temperature. In those early days, of knowledge also sometimes negative effects of elevated [CO2] were reported. However, in almost all cases negative effects of any kind could be exposed as artefacts, due to contaminations of the CO2 gas and a whole bunch of other not controlled circumstances. By the way, readers of this manuscript are not informed about the CO2 source.
800 ppm [CO2] is far away from saturation of photosynthesis. Therefore, this reviewer is convinced that negative effects of this concentration cannot be expected from the physiological point of view. In fact, air temperature close to 40 oC, considered separately, can really damage plants because most plants are reaching their optimum at lower temperatures.
The suspicion (of this reviewer) is that the observed negative effects can only be explained by elevated temperature or some other not known methodical problems. In the opinion of this reviewer there is no plausible reason for a negative combinational effect of [CO2] and temperature within the selected setting of both parameters in this study.
A lot of relevant literature is not cited, and methods have some pitfalls. In addition, in ‘Discussion’ the physiological background for the observed effects is not discussed.
Therefore, this reviewer must apologize for the refusal of this manuscript.
Author Response
Thank you for your review. We have updated the manuscript to improve clarity.

Reviewer 3 Report
Comments and Suggestions for Authors
This manuscript reported an interesting and new study which focused on the changes in leaf morphology of various kinds of plant under simulated warming and elevated CO2 concentration. At present it was well written and has a value for publication. Some minor comments are shown at the bottom for the authors' consideration.
1) In abstract, the authors mentioned the combination of chronic warming and elevated carbon dioxide (eCO2). Can the 7-day incubation in this study be considered as a relative long-term response of plant's morphology to simulated climate change?
2) In the M&M section, the warming level was used as 8oC over optimal temperature. The authors would give the reason for this selection of the increased temperature. By the way, the elevated CO2 concentration was 800 ppm, which is far away from recent atmospheric CO2 concentration. The mutations in air temperature and CO2 concentration could to some extent result in different results, compared with field measurements in nature. Provided that there would be two levels for warming (e.g. 4 and 8oC) and CO2 (600 and 800 ppm) concentrations, the authors would obtain more interesting results. Anyway, the authors may give some discussion or suggestion in the discussion section for future studies to further improve the importance of this study.
Author Response
Thank you for your review. We have updated the manuscript based on your comments.

Round 2
Reviewer 1 Report
Comments and Suggestions for Authors
The authors revised the MS by addressing my comments. Before accepting this paper, I have minor comments that the authors should address.
1. The Conclusion section should be provided.
Author Response
A conclusion has been added, as per the reviewers request.